

# Stability of the vaginal, oral, and gut microbiota across pregnancy among African American women: the effect of socioeconomic status and antibiotic exposure

Anne L. Dunlop[1], Anna K. Knight[2], Glen A. Satten[3], Anya J. Cutler[4], Michelle L. Wright[1], Rebecca M. Mitchell[1,5], Timothy D. Read[6], Jennifer Mulle[7], Vicki S. Hertzberg[1], Cherie C. Hill[2], Alicia K. Smith[2] and Elizabeth J. Corwin[1]

[1] Nell Hodgson Woodruff School of Nursing, Emory University, Atlanta, GA, United States of America
[2] Department of Gynecology and Obstetrics, School of Medicine, Emory University, Atlanta, GA, United States of America
[3] Division of Reproductive Health, Centers for Disease Control and Prevention, Atlanta, GA, United States of America
[4] Department of Environmental Sciences, Emory College, Emory University, Atlanta, GA, United States of America
[5] Department of Computer Sciences, Emory University, Atlanta, GA, United States of America
[6] Department of Medicine, Division of Infectious Disease, School of Medicine, Emory University, Atlanta, GA, United States of America
[7] Department of Human Genetics, School of Medicine, Emory University, Atlanta, GA, United States of America

Corresponding author
Anne L. Dunlop, amlang@emory.edu

## ABSTRACT

**Objective**. A growing body of research has investigated the human microbiota and pregnancy outcomes, especially preterm birth. Most studies of the prenatal microbiota have focused on the vagina, with fewer investigating other body sites during pregnancy. Although pregnancy involves profound hormonal, immunological and metabolic changes, few studies have investigated either shifts in microbiota composition across pregnancy at different body sites or variation in composition at any site that may be explained by maternal characteristics. The purpose of this study was to investigate: (1) the stability of the vaginal, oral, and gut microbiota from early (8–14 weeks) through later (24–30 weeks) pregnancy among African American women according to measures of socioeconomic status, accounting for prenatal antibiotic use; (2) whether measures of socioeconomic status are associated with changes in microbiota composition over pregnancy; and (3) whether exposure to prenatal antibiotics mediate any observed associations between measures of socioeconomic status and stability of the vaginal, oral, and gut microbiota across pregnancy.

**Methods**. We used paired vaginal, oral, or gut samples available for 16S rRNA gene sequencing from two time points in pregnancy (8–14 and 24–30 weeks) to compare within-woman changes in measures of alpha diversity (Shannon and Chao1) and beta-diversity (Bray–Curtis dissimilarity) among pregnant African American women ($n = 110$). Multivariable linear regression was used to examine the effect of level of education and prenatal health insurance as explanatory variables for changes in

diversity, considering antibiotic exposure as a mediator, adjusting for age, obstetrical history, and weeks between sampling.

**Results**. For the oral and gut microbiota, there were no significant associations between measures of socioeconomic status or prenatal antibiotic use and change in Shannon or Chao1 diversity. For the vaginal microbiota, low level of education (high school or less) was associated with an increase in Shannon and Chao1 diversity over pregnancy, with minimal attenuation when controlling for prenatal antibiotic use. Conversely, for within-woman Bray–Curtis dissimilarity for early compared to later pregnancy, low level of education and prenatal antibiotics were associated with greater dissimilarity for the oral and gut sites, with minimal attenuation when controlling for prenatal antibiotics, and no difference in dissimilarity for the vaginal site.

**Conclusions**. Measures of maternal socioeconomic status are variably associated with changes in diversity across pregnancy for the vaginal, oral, and gut microbiota, with minimal attenuation by prenatal antibiotic exposure. Studies that evaluate stability of the microbiota across pregnancy in association with health outcomes themselves associated with socioeconomic status (such as preterm birth) should incorporate measures of socioeconomic status to avoid finding spurious relationships.

# INTRODUCTION

The human microbiota refers to the microbial community inhabiting the human body (*Wang et al., 2017*; *Marchesi & Ravel, 2015*). The human microbiota is increasingly recognized to contribute to biological functions that are important across the lifespan (*Cho & Blaser, 2012*). Research supports the role of the microbiota in digestion and metabolism (*Aziz et al., 2013*; *Bouter et al., 2017*; *Nieuwdorp et al., 2014*), protection against infection and programming of the immune system (*Rooks & Garrett, 2016*; *Günther, Josenhans & Wehkamp, 2016*; *McKenney & Kendall, 2016*; *Round & Mazmanian, 2009*; *Khosravi & Mazmanian, 2013*), production of neurotransmitters (*Yano et al., 2015*; *Strandwitz, 2018*), breakdown of xenobiotics (*Patterson & Turnbaugh, 2014*; *Koppel, Rekdal & Balskus, 2017*), mediation of the physical and emotional response to stress (*Sudo, 2014*; *Bailey et al., 2011*; *Foster, Rinaman & Cryan, 2017*), and many other functions. A growing body of research has investigated the human microbiota and pregnancy outcomes, especially preterm birth. Most studies of the prenatal microbiota have focused on the vagina, given the potential for vaginal microorganisms to ascend to the uterus and the known relationship between intrauterine infection and preterm birth (*Agrawal & Hirsch, 2012*; *Mendz, Kaakoush & Quinlivan, 2013*; *Goldenberg et al., 2008*). Comparatively fewer studies have investigated the microbiota of other body sites during pregnancy (*Barak et al., 2003*; *Madianos, Bobetsis & Offenbacher, 2013*; *Eke et al., 2012*; *Han et al., 2010*; *Han et al., 2006*; *León et al., 2007*; *Bearfield et al., 2002*; *Fujiwara et al., 2017*; *Lin et al., 2018*; *Collado et al., 2016*; *Koren et al., 2012*; *Collado et al., 2008*). Although pregnancy involves profound hormonal, immunological and metabolic changes to support the

fetoplacental unit (*Newbern & Freemark, 2011*), there have been few studies investigating either shifts in microbiota composition across pregnancy at these different sites, or variation in composition at any site that may be explained by maternal characteristics and exposures other than race (*Stout et al., 2017*; *Hyman et al., 2014*).

## The prenatal vaginal microbiota

16S rRNA gene sequencing has been used extensively to study the vaginal microbiota in non-pregnant and pregnant women, finding that a reduction in richness (number of taxa) and evenness (distribution of taxa) (*Aagaard et al., 2011*) and an increase in stability (resistance to change) accompany the transition from the non-pregnant to the pregnant state (*Romero et al., 2014*). Most 16S rRNA gene sequencing surveys characterize the vaginal microbiota into community state types (CSTs) defined via hierarchical clustering and consideration of predominant taxa, with communities clustering into five CST: four dominated by *Lactobacillus iners*, *L. crispatus, L. gasseri* or *L. jensenii,* and a fifth with lower proportion of lactic acid producing bacteria and higher proportions of anerobes (*Zhou et al., 2007*; *Ravel et al., 2011*; *MacIntyre et al., 2015*). A consistent finding is that the proportion of women in different CSTs vary by race, with women of African ancestry significantly more likely to have a vaginal CST *not* dominated by *Lactobacillus* (*Zhou et al., 2007*; *Ravel et al., 2011*; *MacIntyre et al., 2015*). Among women whose vaginal microbiota is dominated by *Lactobacillus*, the predominant species also varies by race, with *L. crispatus* more frequently predominating among Caucasian women and *L. iners,* a species that produces less acid than other *Lactobacillus* species, more commonly predominating among African American women (*Hyman et al., 2014*). Because *L. iners* produces less acid than other *Lactobacillus* spp., it is less effective at maintaining the low pH that usually characterizes vaginal eubiosis (*Amabebe & Anumba, 2018*). Notably, little has been published evaluating the factors that contribute to differences in microbiome composition by race/ethnicity, and these publications have focused on the gut microbiota (*Deschasaux et al., 2018*; *Brooks et al., 2018*).

Studies of vaginal microbiota diversity and pregnancy outcomes (especially preterm birth) have been a focus of research, with conflicting findings across and within racial/ethnic groups. Among studies of mostly Caucasians, one study reports an association between low vaginal species diversity and preterm birth (*Hyman et al., 2014*). Another involving women with prior spontaneous preterm birth finds no such association (*Kindinger et al., 2017*). A third finds that a high-diversity and *Lactobacillus*-poor vaginal microbiota increases the risk of preterm birth relative to one that is low-diversity and *Lactobacillus*-dominated (*DiGiulio et al., 2015*). A small study of nulliparous African American women reports a non-significant association between lower vaginal microbiota diversity and preterm birth (*Nelson et al., 2016*).

More recent studies have focused on shifts in the vaginal microbiota across pregnancy and risk of preterm birth, again with conflicting findings (*Stout et al., 2017*; *Hyman et al., 2014*). A study of mostly Caucasian women found no association between change in vaginal microbiota composition across trimesters and birth outcome (*Hyman et al., 2014*). In contrast, a mostly African American cohort study found that microbial richness,

diversity, and evenness decreased from the first to second trimester for women with preterm birth but remained stable for women with term birth (*Stout et al., 2017*). The discordance in findings may reflect differences in study sample (most are small in size, lack racial and socioeconomic diversity, and are heterogeneous in birth outcome definitions) and methods across studies. The confounding of population and methodological differences across studies makes it difficult to discern whether observed differences in the microbiome by race/ethnicity are real or primarily attributable to differences in confounders such as socioeconomic status by race/ethnicity, which might influence nutrition and stress exposures that themselves affect microbiota composition.

## The prenatal oral microbiota

The bacterial composition of the oral cavity during pregnancy is of interest given the increased risk of periodontal disease during pregnancy (*Barak et al., 2003*) and the link between periodontal disease and both preterm birth and low birth weight (*Madianos, Bobetsis & Offenbacher, 2013*). Of note is the disproportionate occurrence of periodontal disease among African Americans compared to Caucasians, and among individuals with lower income and educational attainment (*Eke et al., 2012*). A few small studies have linked the composition of the oral microbiota to adverse birth outcomes, including two case reports, one linking the presence of intrauterine *Fusobacterium nucleatum* to stillbirth (*Han et al., 2010*) and another linking the presence of intrauterine *Bergeyella* sp. to preterm birth (*Han et al., 2006*) in the presence of pregnancy-associated periodontal disease. One small study linked the presence of *Porphyromonas gingivalis* in gingival samples to microbial invasion of the amniotic cavity in association with preterm labor; *León et al. (2007)* and another small study found that *Streptococcus* spp. and *F. nucleatum* present in the amniotic fluid of women undergoing elective Caesarean section could also be cultured from dental plaque (*Bearfield et al., 2002*).

Two studies have examined the shift in the oral microbiota from the non-pregnant to the pregnant state. One of these compared oral microorganisms identified via culture and polymerase chain reaction among non-pregnant women and pregnant women, finding that the total cultivable microbial counts increase significantly in pregnant compared to non-pregnant women, as well as finding an increase in the incidence of periodontal pathogens (including *Porphyromonas gingivalis* and *Aggregatibacter actinomycetemcomitans*) in the gingival sulcus during early and middle pregnancy compared to the non-pregnant group (*Fujiwara et al., 2017*). Another study using 16S rRNA gene sequencing of supragingival samples from pregnant and non-pregnant women finding the Shannon diversity of pregnant women to be significantly higher than that of non-pregnant women with *Neisseria, Porphyromonas,* and *Treponema* more abundant in the pregnant group, while Streptococcus and Veillonella were more abundant in the non-pregnant group (*Lin et al., 2018*). Few studies have examined shifts in the oral microbiota over the course of pregnancy. One study using 16S rRNA gene sequencing of microbial samples collected weekly from the saliva and the gum in 49 pregnant women, identified high cross-gestational stability, with no significant change in alpha-diversity or beta-diversity across gestation, and no link between specific organisms and adverse outcomes (*DiGiulio et al., 2015*).

## The prenatal gut microbiota

Both gut and oral microorganisms have been reported in the amniotic fluid and placenta of women, including among women experiencing term and preterm birth (*Collado et al., 2016*; *Prince et al., 2016*). Since then, other studies have attributed such findings to laboratory and/or reagent contamination as the bacterial sequences of placental samples could not be distinguished from the contamination background (*Bushman, 2019*; *Theis et al., 2019*; *Lauder et al., 2016*) and studies conducted among healthy, full term deliveries found no placental microbiome (*Lim, Rodriguez & Holtz, 2018*; *Leiby et al., 2018*).

To date, several studies using 16S rRNA gene sequencing have been published on the gut microbiota during pregnancy, with different findings of stability across pregnancy (*Koren et al., 2012*; *Collado et al., 2008*; *DiGiulio et al., 2015*). One study involving repeated longitudinal sampling across gestational weeks identified high cross-gestational stability, with no significant change in alpha-diversity or beta-diversity over gestational weeks of the pregnancy (*DiGiulio et al., 2015*). Another that involved sampling in the first and third trimesters found an expansion of diversity, with an overall increase in *Proteobacteria* and *Actinobacteria*, and reduced richness from the first to the third trimesters; however, microbiome gene composition was constant between trimesters (*Koren et al., 2012*). A third study compared the gut microbiota in overweight and normal weight women in the first and third trimesters of pregnancy using fluorescent *in situ* hybridization (FISH) and quantitative real-time polymerase chain reaction (qPCR), finding increasing microbial counts from the first to the third trimester overall and increasing abundance of *Bacteroides* and *Staphyloccoccus* among women who were overweight compared to normal weight (*Collado et al., 2008*). This same study noted that the prenatal gut microbiota may be influenced by pre-pregnancy weight as well as weight gain over the course of the pregnancy (*Collado et al., 2008*). In a murine model, pregnancy-related changes in the maternal gut microbiota were found to depend upon the mother's periconceptional diet (*Gohir et al., 2015*).

## Sociodemographic features and the prenatal microbiota

Given the persistent and long-standing racial disparities in poor birth outcomes, such as preterm birth (*Behrman & Butler, 2007*), studies have focused on both variation in microbiota composition by race and the potential role of such differences in explaining disparities in birth outcomes (*Hyman et al., 2014*; *Zhou et al., 2007*; *Ravel et al., 2011*; *MacIntyre et al., 2015*; *Kindinger et al., 2017*; *DiGiulio et al., 2015*; *Nelson et al., 2016*). In the United States, differences in health outcomes for groups defined by race/ethnicity are oftentimes confounded by inter-group differences in socioeconomic status, which often go unmeasured, unreported, or unaccounted for in analyses (*LaVeist, 2005*; *Smith, 2000*). The extent to which socioeconomic status plays a role in microbiota composition and stability, particularly within pregnancy, has been relatively unexplored. Such research, however, is important for ascertaining whether observed differences in microbiota composition and stability by race are confounded by socioeconomic status.

## Antibiotics and the prenatal microbiome

Antibiotics have both short- and long-term effects on the microbiota, affecting the target pathogen as well as commensal inhabitants (*Jernberg et al., 2010*; *Modi, Collins & Relman, 2014*). Antibiotics account for 80% of all medications prescribed in pregnancy (*Bookstaver et al., 2015*). Few published studies have, however, evaluated the effects of prenatal antibiotics on the maternal or fetal microbiome (*Mueller et al., 2015*; *Stokholm et al., 2014*). It has been demonstrated that prenatal antibiotics alter the vaginal microbiota, which may have later effects on the colonization of the newborn (*Stokholm et al., 2014*) and childhood obesity (*Mueller et al., 2015*). Most of the common conditions that result in antibiotic use in pregnancy include genitourinary infections, such as bacterial vaginosis, which occur at higher rates among African American women compared to women of other race/ethnicity (*Chesson et al., 2012*). As a result, antibiotic use is another potential modifier of observed racial differences in studies of the microbiota and health outcomes.

## Goals of this study

The purpose of this study was to investigate: (1) the stability of the vaginal, oral, and gut microbiota from early (8–14 weeks) through later (24–30 weeks) pregnancy among African American women according to measures of socioeconomic status, accounting for prenatal antibiotic use; (2) whether measures of socioeconomic status are associated with changes in microbiota composition over pregnancy; and (3) whether exposure to prenatal antibiotics mediate any observed associations between measures of socioeconomic status and stability of the vaginal, oral, and gut microbiota across pregnancy. The rationale for restricting the study to African American women is based on a health disparity research framework that recommends that researchers, as a first step to understanding health disparities, look within the high burden group to identify intra-group risk and protective factors (*Rowley et al., 1993*), and the observation that existing studies of the microbiota composition in pregnancy do not consider socioeconomic status and hence are not able to parse the possibly competing effects of race and socioeconomic status.

# MATERIALS & METHODS

## Participants

Participants for this study were drawn from women participating in the Emory University African American Vaginal, Oral, and Gut Microbiome in Pregnancy Study (*Corwin et al., 2017*). African American women were recruited from two hospitals in Atlanta, GA: one a private hospital affiliated with Emory University that provides services for a socioeconomically and educationally diverse group of women, and the second, Grady Hospital, a public facility also staffed by Emory obstetrical faculty, that primarily provides services for low-income women. Inclusion criteria for enrollment into the cohort included: (1) being African American, defined for purposes of this study as being of self-reported Black/African American race and born in the United States; (2) presenting between 8 and 14 weeks gestation (verified by clinical record and/or ultrasound) with a singleton pregnancy; (3) ability to comprehend written and spoken English; (4) age between 18–40 years; (5) absence of a chronic medical condition, as well as absence of chronic use of prescription

medication (verified by prenatal record). Women who developed health conditions or pregnancy complications, including those that required prescription medications were retained in the study and these exposures and their gestational age of occurrence were recorded. The present study included participants who had paired vaginal, oral, or gut samples available for 16S rRNA gene sequencing from both sample collection time points. The research protocol was reviewed and approved by the Emory University Institutional Review Board (protocol number 68441); all participants provided written informed consent.

## Data collection

Data collection has been described in detail previously (*Corwin et al., 2017*), including biological samples, clinical and questionnaire data at two points during pregnancy (at prenatal care visits occurring at 8–14 and 24–30 weeks); and clinical data (from the medical record) post-delivery. Items relevant to this study are described below.

*Sociodemographic survey* based on maternal self-report and prenatal administrative record review was used to ascertain maternal age, years of education (collected as a four-level variable; categorized as high school or less vs. some college or more in analyses), and prenatal health insurance type (categorized as Medicaid vs. private insurance). To be eligible for Medicaid coverage during pregnancy in Georgia, women must have a household income below 200 percent of the federal poverty level.

*Medical chart abstraction* using a standardized chart abstraction tool was undertaken to determine gestational age at time of sample collection, any diagnoses of infection, and/or prescription of systemic or oral antibiotics (with antibacterial actions) by comparing the date of these occurrences to the estimated date of confinement based on last menstrual period (LMP) and/or ultrasound before 14 weeks' gestation according to standard clinical criteria (*Obstetricians ACo, Gynecologists, 2014*). Participants were coded as having been exposed to antibiotics before Visit 1 if they were prescribed an antibiotic in the four weeks prior to Visit 1 and were considered to have been exposed to antibiotics between Visit 1 and Visit 2 if they were prescribed an antibiotic at any time between the two visits. Chart abstraction was also used to ascertain obstetrical history, in terms of whether the woman had any previous term births or any previous preterm births (considered as two separate categorical variables).

## Vaginal, oral, and rectal swabs

Participants were provided verbal and pictorial instructions explaining how to obtain self-collected vaginal, oral, and rectal swabs. The vaginal collection involved sampling the midportion of the vaginal vault (3–4 inches from introitus). The oral collection involved sampling the tongue, hard palate, and gum line. The rectal sample involved sampling the rectal vault 1 inch beyond the anal sphincter. Consistent with the protocols of the Human Microbiome Project (*McInnes & Cutting, 2010*), the sampling at each body site used a Sterile Catch-All™ Sample Collection Swab (Epicentre Biotechnologies, Madison WI) that was immediately handed to the study coordinator for placement in MoBio bead tubes (MoBio Laboratories, Inc.,) that were frozen upright on ice until transported to

the lab, where they were stored at −80 °C until DNA extraction. Studies support that vaginal self-collection swabs sample the same microbial diversity as physician-collected swabs of the mid-vagina and have high overall morphotype-specific validity compared with provider-collected swabs (*Forney et al., 2010*).

### DNA extraction, 16S rRNA gene library prepration and sequencing

DNA was extracted from participant swab samples using the MoBio isolation Kit in line with the HMP Standard Operating Protocol. DNA quantification based on a threshold of 5 ug/nL was used to identify samples that were borderline in terms of DNA yield; in cases that were borderline, DNA quality was assessed on a 2% agarose gel and quantitated with the Broad Range Quant-It kit from ThemoFisher Scientific (Q33130). Participant samples with DNA visible on the gel were sequenced as were no-template controls that contained all assay components except for DNA, which were used to verify the lack of contamination across reagents and samples. Microbial diversity was characterized by DNA sequence variation of the 16S rRNA gene. The variable V3 and V4 regions of the 16S rRNA gene were amplified and tailed using target specific primers, Illumina sequencing adapters, and barcodes according to the Illumina 16S Metagenomic Sequencing Library Preparation guide (version 15044223-b) (*Caporaso et al., 2011*). Each sample DNA was amplified in duplicate, to control for variation due to random PCR amplification artifacts. Quantified libraries were pooled and sequenced at 10pM loading density with 20% PhiX spike-in (FC-110-3001) on an Illumina MiSeq using v3 600 cycle MiSeq Reagent chemistry (Illumina, catalog # MS-102-3003), generating approximately 20M PE300 reads (*Caporaso et al., 2011*). For each run, more than 10 million high quality paired reads ($Q$ score > 30 at each base) were identified for an average number of reads of greater than 50,000 per sample (average 58,046, minimum 2,027, maximum 159,296), after control DNA removal. Metadata on each sample was stored in a local database compliant with the MIMS (Minimal Information about a Metagenome Sequence) ontology.

### Data quality control and bioinformatic processing

Raw fastq files were imported into Qiime2 version 2017.12 and underwent quality control and denoising with dada2 (*Callahan et al., 2016*; *McDonald et al., 2012*). 22 base pairs were trimmed from both the left and right ends of each sequence. Each read was truncated at 250 base pairs. Each run was processed separately and merged after quality control. Samples with fewer than 2,027 reads after quality control were excluded from further analyses. ASVs were classified using the 11.5 release of the Ribosomal Database Project (RDP) classifier with a minimum bootstrap confidence of 80%, implemented in dada2 (*Callahan et al., 2016*; *Maidak et al., 1994*). Feature tables of amplicon sequence variants (ASVs) were exported and downstream analyses were performed using the Phyloseq R package in R version 3.4.0 (*McMurdie & Holmes, 2013*).

### Statistical analysis

We summarized characteristics of participating women descriptively and compared differences in the characteristics of women providing paired samples across the various body sites using Chi-square or $t$-test, as appropriate. For each body site, we also compared

women providing paired samples to women not providing paired samples to ensure that the subset of women providing paired samples for a given body site was not a biased set of the participating women. Shannon and Chao1 measures of alpha-diversity were calculated for each body site for both sample collection time points (Visit 1 and Visit 2). Diversity measures were based on the average of 1,000 rarefactions to the minimum library size of 2,027. ASVs that were unassigned at the genus level were excluded from diversity calculations. We calculated the change in Shannon and Chao1 diversities by subtracting the calculated measure for Visit 2 data from the calculated measure for Visit 1 data. The Bray–Curtis dissimilarity index, a measure of beta-diversity used to quantify the compositional dissimilarity between two sites based on counts, was calculated for paired Visit 1 and Visit 2 samples from the same body site for each woman based on ASV frequencies. The Bray–Curtis Dissimilarity index was selected for its ease of interpretability (it is bounded by 0 and 1, with 0 indicating that the two samples have the same composition and 1 indicating that the two samples do not share any taxa) and its avoidance in making assumptions about evolutionary relationships (*Lucas et al., 2017*). We generated box plots to compare the distribution of the change in measures of alpha- and beta-diversity from Visit 1 to Visit 2, and compared these distributions across body sites. Because samples from both time points were run on the same plate for each body site for data from all women included in the analyses reported here, no adjustment for batch or plate was required.

To evaluate the effect of the variables of interest on the stability of the microbiota from Visit 1 to Visit 2 for a given woman (i.e., the within-woman changes in measures of alpha- and beta-diversity from Visit 1 to Visit 2), we used multivariate linear regression. For each body site we performed univariate analyses using maternal level of education and prenatal health insurance as the principal explanatory variables and change in Chao1 and Shannon diversity and the Bray–Curtis dissimilarity index as the outcome variables (in separate models); we then performed multivariate analyses that additionally controlled for age, obstetrical history (any previous term birth, any previous preterm birth), and weeks between sampling and prenatal antibiotic exposure before Visit 1 or between Visit 1 and 2.

To evaluate whether women who experience the same change in ASV from Visit 1 to Visit 2 for a particular body site share the same co-variates, we first calculated the difference in ASV frequencies between the two visits, and then calculated the Manhattan (L1) distance between these frequency differences for each pair of women in the study for each body site. We then performed a PERMANOVA (between-woman analysis) to assess the extent to which compositional shifts between Visit 1 and Visit 2, as captured by the Manhattan distance, were associated with measures of socioeconomic status and/or antibiotic exposure. The Manhattan distance was chosen for this aspect of our analysis as it is proportional to the Bray–Curtis dissimilarity index when ASV frequencies are used, but the Manhattan distance allows for non-negative values (unlike the Bray–Curtis), which occur when considering differences in ASV frequencies (*Lucas et al., 2017*).

## RESULTS

### Participants

There were a total of 122 African American women in the cohort who had paired (Visit 1 and Visit 2) samples for 16S rRNA gene sequencing for at least one body site for inclusion in this study (Table 1); of these, after quality control, 110 had paired vaginal samples, 97 had paired oral samples, and 69 had paired gut samples. There were no significant differences in the demographic characteristics of women providing paired vaginal, oral, or gut samples. Table 1 also shows the difference in summary measures for characteristics for women who did and did not contribute paired samples from each body site. Women who contributed paired vaginal samples were significantly more likely to have had a prior term birth than were women who did not; women who contributed paired gut samples were significantly more likely to have graduated from college than were those who did not. The mean age of the women studied was approximately 24 years, and approximately 78% had Medicaid as prenatal health insurance. Approximately 15% of the sample had received antibiotics within four weeks prior to Visit 1 and approximately 22% received antibiotics between Visit 1 and Visit 2. The most common indications for receiving antibiotics before Visit 1 or between Visit 1 and Visit 2 were bacterial vaginosis, chlamydia cervicitis and/or vaginitis, and urinary tract infection or asymptomatic bacteriuria; the three most common antibiotics to which women were exposed included oral metronidazole (5–7 day course), oral azithromycin (single dose), and oral nitrofurantoin (5–7 days course). Exposure to antibiotics before Visit 1 was not significantly different according to education level ($p = 0.98$) or type of health insurance ($p = 0.58$). However, exposure to antibiotics between Visit 1 and Visit 2 was higher among women with low education (high school or less vs. some college or higher) ($p = 0.02$) and higher among women with Medicaid compared to private insurance ($p = 0.02$) (Table 1).

### 16S rRNA sequencing data

For the 122 individuals with paired samples from the vaginal, oral, or gut body site that were included in this study, 26 samples (and their pairs) were removed as they represented statistical outliers in relation to the other samples (four samples with greater than 175,000 reads and 22 samples with fewer than 2,000 reads). From these 552 samples, 37,165,746 reads grouped into 11,450 ASVs remained after filtering out singletons and ASVs present in only one sample. After removing reads that could not be classified to the level of genus, sequence data representing 5,090 ASVs comprised of 32,406,428 reads remained for analysis after quality control. For samples run in duplicate ($N = 44$), the sample with the higher read count was retained and the other dropped before creation of the biome table; no further analyses were done on the data that were dropped.

### Change in alpha-diversity of vaginal, oral, and gut microbiota across pregnancy

The distribution of the within-woman change in measures of alpha-diversity of the microbiota from Visit 1 to Visit 2 at the three body sites is shown in Fig. 1. For all three body sites, the difference in Chao1 diversity (richness) and Shannon diversity (evenness)

**Table 1  Characteristics of the Study Sample (women with paired samples).**

| Characteristic | All subjects N = 122 | Vaginal sample N = 110 | | Oral sample N = 97 | | Gut sample N = 69 | |
|---|---|---|---|---|---|---|---|
| | | Difference[1] | p-value[2] | Difference[1] | p-value[2] | Difference[1] | p-value[2] |
| Age, years (mean ± sd) | 24.07 ± 4.4 | −1.6 | 0.17 | 2.03 | 0.13 | −.44 | 0.89 |
| Prenatal health insurance, n (%) | | | | | | | |
| Medicaid | 94 (77) | −11.5 | 0.47 | −1.3 | 0.99 | .55 | .051 |
| Private | 28 (23) | 11.5 | 0.47 | 1.3 | 0.99 | −.55 | .051 |
| Education level | | | | | | | |
| Less than high school | 19 (16) | 1.2 | 0.99 | −4.5 | 0.65 | 2.5 | 0.99 |
| High school or GED | 52 (43) | 1.1 | 0.99 | 1.7 | 0.74 | −12.0 | 0.51 |
| Some college | 36 (30) | 14.2 | 0.51 | −6.9 | 0.72 | 11.2 | 0.99 |
| College graduate | 15 (13) | 14.1 | 0.17 | 9.7 | 0.99 | −1.7 | **0.01** |
| Marital status, n (%) | | | | | | | |
| Married or cohabiting | 66 (54) | 23.0 | 0.14 | 7.8 | 0.18 | −1.1 | 0.99 |
| Obstetrical history, n (%) | | | | | | | |
| Prior term birth | 59 (49) | −63.3 | **0.04** | 17.9 | 0.10 | 3.0 | 0.68 |
| Prior preterm birth | 14 (11) | 15.5 | 0.71 | −12.5 | 0.69 | −8.0 | 0.99 |
| Exposure to antibiotics, n (%) | | | | | | | |
| Prior to study visit 1 | 26 (22) | −14.4 | 0.46 | −6.7 | 0.69 | −11.0 | 0.99 |
| Between study visit 1 and 2 | 34 (28) | 6.1 | 0.74 | 0.16 | 0.46 | 0.77 | 0.99 |

**Notes.**
[1] Difference in summary measure for participants with paired samples from the body site compared to those without paired samples from the site.
[2] p-value for t-test (for continuous variables) or Fishers exact test (for categorical values).
The bold styling indicates a p-value that is statistically significant at $\alpha < 0.05$.

from Visit 1 to Visit 2 appears to be approximately centered around zero, reflecting no difference in Chao1 and Shannon diversity for the vaginal, oral, or gut microbiome across pregnancy for the group overall.

Results of univariate and multivariate analyses of the within-woman change in alpha-diversity from Visit 1 to Visit 2 are given in Table 2 (Shannon) and 3 (Chao1). In univariate analysis, having a low level of education (high school or less) was associated with an increase in vaginal Shannon diversity from Visit 1 to Visit 2, and this association was not attenuated by controlling for age, history of term birth, history of preterm birth, or weeks between sampling (multivariate model 1); nor was the association attenuated when additionally controlling for antibiotic exposure in the month prior to Visit 1 or between Visit 1 and Visit 2 (multivariate model 2). No significant associations were observed between change in Shannon diversity across pregnancy and measures of socioeconomic status and prenatal antibiotic exposure for the oral and gut sites (Table 2).

In univariate analysis, low level of education was not significantly associated with change in Chao1 diversity. However, in multivariate modeling, low level of education was significantly associated with an increase in vaginal Chao1 diversity across pregnancy, when controlling for age, obstetrical history (history of term birth, history of preterm birth), and weeks between sampling (multivariate model 1); this association was mildly attenuated when additionally controlling for antibiotic exposure (multivariate model 2).

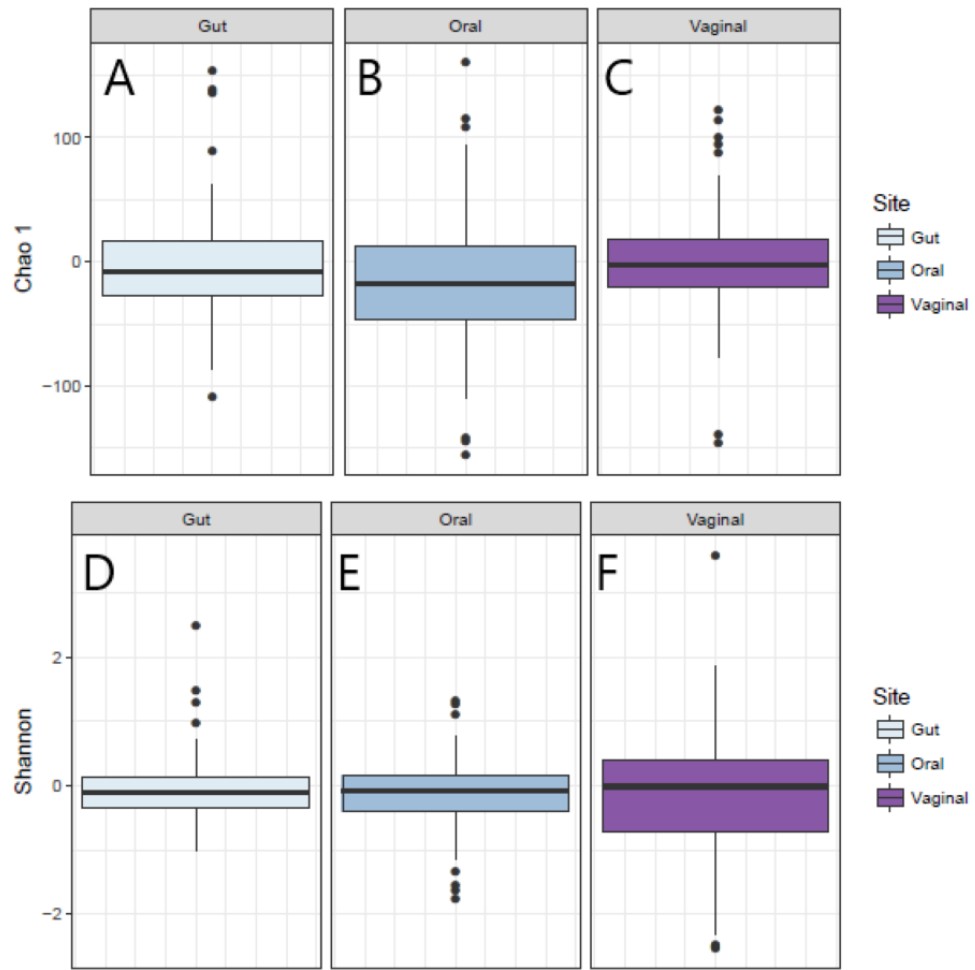

**Figure 1 Change in Shannon and Chao1 measures for Visit 1 and Visit 2 by body site.** (A) Distribution of change in Chao1 diversity from Visit 1 to Visit 2 for gut microbiota. (B) Distribution of change in Chao1 diversity from Visit 1 to Visit 2 for oral microbiota. (C) Distribution of change in Chao1 diversity from Visit 1 to Visit 2 for vaginal microbiota. (D) Distribution of change in Shannon diversity from Visit 1 to Visit 2 for gut microbiota. (E) Distribution of change in Shannon diversity from Visit 1 to Visit 2 for oral microbiota. (F) Distribution of change in Shannon diversity from Visit 1 to Visit 2 for vaginal microbiota.

No significant relationships were observed between change in Chao1 and measures of socioeconomic status and antibiotic exposure for the oral and gut sites (Table 3).

## Change in beta-diversity of vaginal, oral, and gut microbiota across pregnancy

The distribution of the within-woman change in microbiota composition from Visit 1 to Visit 2 as measured by the Bray–Curtis dissimilarity index is given in Fig. 2. The median within-woman Bray–Curtis dissimilarity index was approximately 0.5, indicating on average half of the probability mass had shifted between the two visits. This finding was similar for the vaginal, oral, and gut body sites; however, the interquartile range was

**Table 2  Change in Shannon diversity, according to socioeconomic status and antibiotic exposure.**

| Body site/exposure | Univariate association | | Multivariate association[1] | | Multivariate association[2] | |
|---|---|---|---|---|---|---|
| | Coefficent | *p*-value | Coefficent | *p*-value | Coefficent | *p*-value |
| **VAGINAL** | | | | | | |
| Level of Education[A] | **0.48** | **0.02** | **0.50** | **0.03** | **0.55** | **0.02** |
| Prenatal Insurance[B] | 0.25 | 0.31 | 0.19 | 0.51 | 0.26 | 0.43 |
| Marital/Cohabiting[C] | 0.10 | 0.61 | 0.05 | 0.81 | 0.08 | 0.71 |
| Antibiotics prior to Visit 1 | −0.14 | 0.57 | −0.16 | 0.52 | N/A | N/A |
| Antibiotics between Visit 1 & 2 | −0.17 | 0.46 | −0.19 | 0.41 | N/A | N/A |
| **ORAL** | | | | | | |
| Level of Education[A] | 0.03 | 0.77 | −0.01 | 0.94 | 0.001 | 0.99 |
| Prenatal Insurance[B] | −0.13 | 0.34 | −0.20 | 0.19 | −0.19 | 0.21 |
| Marital/Cohabiting[C] | 0.08 | 0.49 | 0.05 | 0.66 | 0.06 | 0.75 |
| Antibiotics prior to Visit 1 | 0.07 | 0.62 | 0.05 | 0.75 | N/A | N/A |
| Antibiotics between Visit 1 & 2 | −0.03 | 0.84 | −0.05 | 0.73 | N/A | N/A |
| **GUT** | | | | | | |
| Level of Education | 0.07 | 0.63 | 0.05 | 0.74 | 0.05 | 0.78 |
| Prenatal Insurance | −0.03 | 0.84 | −0.09 | 0.61 | −0.10 | 0.58 |
| Marital/Cohabiting | 0.03 | 0.80 | 0.07 | 0.60 | 0.08 | 0.60 |
| Antibiotics prior to Visit 1 | 0.12 | 0.42 | 0.13 | 0.40 | N/A | N/A |
| Antibiotics between Visit 1 & 2 | −0.02 | 0.90 | 0.01 | 0.98 | N/A | N/A |

**Notes.**

[1] Model controlling for age, any previous term birth, any previous preterm birth, weeks between sampling.

[2] Model controlling for age, any previous term birth, any previous preterm birth, weeks between sampling, and antibiotics.

[A] Education coded as 1 for high school or lower, 0 for some college or higher.

[B] Prenatal insurance coded as 1 for Medicaid, 0 for Private.

[C] Marital/Co-habiting coded as 1 for Single or Not cohabiting, 0 for Married or Cohabiting.

The bold styling indicates a *p*-value that is statistically significant at $\alpha < 0.05$.

substantially wider for the vaginal compared to the oral and gut sites, suggesting greater variability in intra-individual change for the vaginal site.

Univariate and multivariate analytic results for the within-woman longitudinal stability from Visit 1 to Visit 2 for each body site, as captured by the Bray–Curtis dissimilarity index, are given in Table 4. For the vaginal microbiota, measures of socioeconomic status (level of education, prenatal insurance) and antibiotic exposure were not associated with the Bray–Curtis dissimilarity in univariate or multivariate modeling. For the oral microbiota, having a low level of education and receipt of antibiotics between visits were associated with greater Bray–Curtis dissimilarity, with some attenuation of the effect of education when additionally controlling for prenatal antibiotics. For the gut microbiota, having a low level of education and receiving antibiotics between visits was also associated with greater Bray–Curtis dissimilarity, with elimination of the effect of education when controlling for age, obstetrical history, and weeks between sampling (multivariate model 1) and when controlling for age, obstetrical history, weeks between sampling, and antibiotics (multivariate model 2).

Univariate and multivariate analytic results for the association between socioeconomic variables with the direction of compositional change, as captured by the Manhattan distance

**Table 3  Change in Chao 1 diversity, according to socioeconomic status and antibiotic exposure.**

| Body site/exposure | Univariate association | | Multivariate association[1] | | Multivariate association[2] | |
|---|---|---|---|---|---|---|
| | Coefficent | p-value | Coefficent | p-value | Coefficent | p-value |
| **VAGINAL** | | | | | | |
| Level of Education[A] | 13.3 | 0.10 | **18.2** | **0.04** | 17.6 | 0.05 |
| Prenatal Insurance[B] | 2.0 | 0.83 | 3.4 | 0.75 | 3.0 | 0.78 |
| Marital/Cohabiting[C] | 9.8 | 0.22 | 8.5 | 0.31 | 7.8 | 0.36 |
| Antibiotics prior to Visit 1 | −14.9 | 0.12 | −13.6 | 0.16 | N/A | N/A |
| Antibiotics between Visit 1 and 2 | 3.0 | 0.74 | 3.6 | 0.69 | N/A | N/A |
| **ORAL** | | | | | | |
| Level of Education[A] | −1.4 | 0.90 | −5.2 | 0.67 | −5.0 | 0.69 |
| Prenatal Insurance[B] | −16.6 | 0.19 | −23.9 | 0.09 | −23.9 | 0.10 |
| Marital/Cohabiting[C] | 9.8 | 0.36 | 7.5 | 0.51 | 7.8 | 0.50 |
| Antibiotics prior to Visit 1 | −4.6 | 0.72 | −5.9 | 0.65 | N/A | N/A |
| Antibiotics between Visit 1 and 2 | −1.1 | 0.93 | −2.5 | 0.84 | N/A | N/A |
| **GUT** | | | | | | |
| Level of Education[A] | 6.1 | 0.61 | 6.5 | 0.66 | 7.5 | 0.62 |
| Prenatal Insurance[B] | 3.3 | 0.81 | 1.2 | 0.94 | 1.9 | 0.91 |
| Marital/Cohabiting[C] | −9.4 | 0.42 | −7.1 | 0.57 | −5.9 | 0.65 |
| Antibiotics prior to Visit 1 | 6.5 | 0.62 | 7.2 | 0.59 | N/A | N/A |
| Antibiotics between Visit 1 and 2 | −8.5 | 0.51 | −6.6 | 0.62 | N/A | N/A |

Notes.
[1] Model controlling for age, any previous term birth, any previous preterm birth, weeks between sampling.
[2] Model controlling for age, any previous term birth, any previous preterm birth, weeks between sampling, and antibiotics.
[A] Education coded as 1 for high school or lower, 0 for some college or higher.
[B] Prenatal insurance coded as 1 for Medicaid, 0 for Private.
[C] Marital/Co-habiting coded as 1 for Single or Not cohabiting, 0 for Married or Cohabiting.
 The bold styling indicates a p-value that is statistically significant at $\alpha < 0.05$.

on a matrix of differences in ASV reads between Visit 1 and Visit 2, for each body site are given in Table 5. For the vaginal site, level of education, insurance type, and antibiotic exposure were not associated with inter-individual compositional change between Visits 1 and 2. However, for the oral and gut sites, prenatal insurance type was associated with the direction of compositional change from Visit 1 to Visit 2. Prenatal antibiotic use was not associated with, nor did it substantially attenuate the effect of prenatal insurance type, on the direction of change in oral and gut composition across pregnancy.

## DISCUSSION

Taken together, the findings from our research, in conjunction with the current small body of literature, support the contention that variables linked to socioeconomic status are associated with changes in microbiota composition and this association is minimally attenuated by prenatal antibiotic exposure. This suggests that ignoring variables linked to socioeconomic status when assessing the association between the microbiota and health outcomes may lead to spurious associations. Given the strong relationships between race/ethnicity and socioeconomic status in the United States, and the known positive associations between minority race and low socioeconomic status and adverse birth

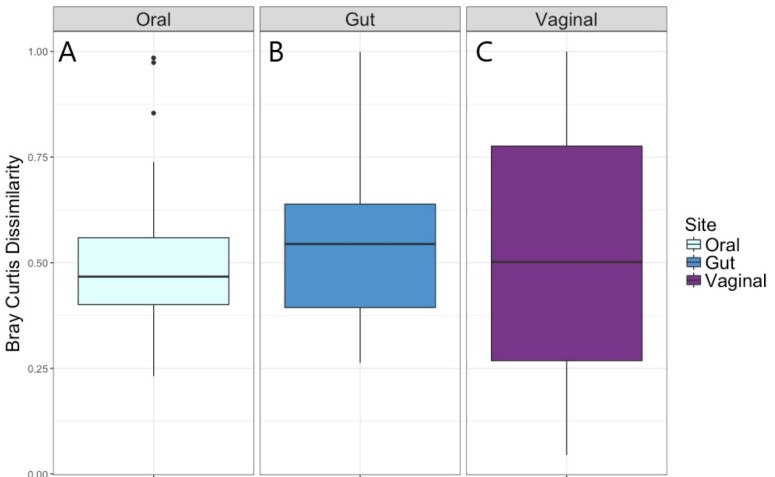

**Figure 2** **Bray–Curtis dissimilarity measure for Visit 1 and Visit 2 by body site.** (A) Distribution of Bray–Curtis dissimilarity from Visit 1 to Visit 2 for oral microbiota. (B) Distribution of Bray–Curtis dissimilarity from Visit 1 to Visit 2 for gut microbiota. (C) Distribution of Bray–Curtis dissimilarity from Visit 1 to Visit 2 for vaginal microbiota.

outcomes, such as preterm birth (*Behrman & Butler, 2007*), the inclusion of measures of socioeconomic status in determining potential differences in the relationship between the microbiota and preterm birth may be especially important.

Very few studies have examined the association between measures of socioeconomic status and the composition and stability of the microbiota across pregnancy. For the vaginal microbiota, one study based on a sample from the Vaginal Human Microbiome Project at Virginia Commonwealth University compared the microbiota of vaginal samples from 1,268 African American women and 416 women of European ancestry, finding significant differences in the vaginal microbiota of the two ethnic groups and identified several taxa relevant to these differences; however, only ethnicity, pregnancy, and alcohol use correlated significantly with the relative abundance of bacterial-vaginosis-associated species. While household income was significantly correlated with ethnicity, it itself was not significantly associated with the relative abundance of bacterial-vaginosis-associated species (*Fettweis et al., 2014*).

We are aware of only two studies that have considered the effect of socioeconomic status on the gut microbiota. In one US study of 44 healthy adult volunteers, investigators found that in adjusted analyses, census tract measures of neighborhood socioeconomic status explained 11–22% of the variability in diversity indicators, and that residence in neighborhoods of higher socioeconomic status was associated with greater abundance of *Bacteroides* and lower abundance of *Prevotella* (*Miller et al., 2016*). A UK twin study likewise found a greater abundance of *Bacteroides* with higher socioeconomic status and, furthermore, that diet as a mediating factor does not completely explain variance of alpha and beta diversity associated with socioeconomic status (*Bowyer et al., 2019*).

**Table 4** Bray–Curtis dissimilarity between Visit 1 and Visit 2, according to socioeconomic status and antibiotic exposures for vaginal, oral, and gut body sites.

| Body site/exposure | Univariate association | | Multivariate association[1] | | Multivariate association[2] | |
|---|---|---|---|---|---|---|
| | Coefficent | p-value | Coefficent | p-value | Coefficent | p-value |
| **VAGINAL** | | | | | | |
| Level of Education[A] | 0.046 | 0.41 | 0.022 | 0.72 | 0.015 | 0.81 |
| Prenatal Insurance[B] | 0.125 | 0.06 | 0.101 | 0.17 | 0.094 | 0.21 |
| Marital/Cohabiting | 0.080 | 0.15 | 0.070 | 0.23 | 0.065 | 0.27 |
| Antibiotics prior to Visit 1 | 0.054 | 0.41 | 0.050 | 0.74 | N/A | N/A |
| Antibiotics between Visit 1 and 2 | 0.056 | 0.37 | 0.051 | 0.41 | N/A | N/A |
| **ORAL** | | | | | | |
| Level of Education[A] | **0.068** | **0.01** | **0.077** | **0.015** | 0.060 | 0.05 |
| Prenatal Insurance[B] | 0.027 | 0.42 | 0.020 | 0.58 | 0.010 | 0.79 |
| Marital/Cohabiting[C] | 0.030 | 0.35 | 0.029 | 0.32 | 0.019 | 0.52 |
| Antibiotics prior to Visit 1 | 0.012 | 0.73 | 0.008 | 0.82 | N/A | N/A |
| Antibiotics between Visit 1 and 2 | **0.083** | **0.006** | **0.083** | **0.008** | N/A | N/A |
| **GUT** | | | | | | |
| Level of Education[A] | **0.098** | **0.03** | 0.078 | 0.15 | 0.058 | 0.28 |
| Prenatal Insurance[B] | 0.079 | 0.12 | 0.038 | 0.51 | 0.020 | 0.73 |
| Marital/Cohabiting[C] | **0.140** | **0.001** | **0.130** | **0.004** | **0.11** | **0.02** |
| Antibiotics prior to Visit 1 | 0.031 | 0.53 | 0.022 | 0.66 | N/A | N/A |
| Antibiotics between Visit 1 and 2 | **0.120** | **0.009** | **0.110** | **0.021** | N/A | N/A |

**Notes.**

[1] Model controlling for age, any previous term birth, any previous preterm birth, weeks between sampling.

[2] Model controlling for age, any previous term birth, any previous preterm birth, weeks between sampling, and antibiotics.

[A] Education coded as 1 for high school or lower, 0 for some college or higher.

[B] Prenatal insurance coded as 1 for Medicaid, 0 for Private.

[C] Marital/Co-habiting coded as 1 for Single or Not cohabiting, 0 for Married or Cohabiting.

The bold styling indicates a p-value that is statistically significant at $\alpha < 0.05$.

Two studies have considered the effect of socioeconomic status on the composition of the oral microbiota, but only in non-pregnant adult populations. In one of these studies, data from a representative sample of 296 adult residents of New York City examined associations between the structure and diversity of the oral microbiota and socioeconomic variables including age, gender, income, education, nativity, and race/ethnicity. This study found that 79 operational taxonomic units were differentially abundant by measures of socioeconomic status: 52 by age group, 27 by race/ethnicity, 14 by income, 14 by education, 12 by nativity, and 5 by gender (*Renson et al., 2017*). In the second study, data from 292 participants enrolled in the Danish Health Examination Survey found distinct clustering of the oral microbiota by measures of socioeconomic status (*Belstrøm et al., 2014*).

While it is known that antibiotics induce changes in the gut microbiota (*Becattini, Taur & Pamer, 2016*), existing studies have not evaluated the effect of antibiotic exposures on the composition and stability of the vaginal, oral, and gut microbiota in pregnancy. Further, no studies have considered the role of antibiotic exposures on the changes in the microbiota composition across pregnancy, and how this may affect the relationship between the microbiome, microbiome changes, and preterm birth.

**Table 5** PERMANOVA on Manhattan distance between Visit 1 and Visit 2, according to socioeconomic status and antibiotic exposures for vaginal, oral, and gut body sites.

| Body site/exposure | Univariate model | | Multivariate model[1] | | Multivariate model[2] | |
|---|---|---|---|---|---|---|
| | $R^2$ | *p*-value | $R^2$ | *p*-value | $R^2$ | *p*-value |
| **VAGINAL** | | | | | | |
| Level of Education[A] | 0.011 | 0.28 | 0.011 | 0.29 | 0.011 | 0.29 |
| Prenatal Insurance[B] | 0.014 | 0.13 | 0.014 | 0.15 | 0.014 | 0.15 |
| Marital/Cohabiting[C] | 0.006 | 0.75 | 0.006 | 0.75 | 0.006 | 0.75 |
| Antibiotics prior to Visit 1 | 0.010 | 0.37 | 0.010 | 0.37 | N/A | N/A |
| Antibiotics between Visit 1 and 2 | 0.011 | 0.27 | 0.011 | 0.27 | N/A | N/A |
| **ORAL** | | | | | | |
| Level of Education[A] | 0.013 | 0.14 | 0.013 | 0.16 | 0.013 | 0.14 |
| Prenatal Insurance[B] | **0.018** | **0.01** | **0.018** | **0.01** | **0.018** | **0.009** |
| Marital/Cohabiting[C] | 0.010 | 0.46 | 0010 | 0.45 | 0.010 | 0.44 |
| Antibiotics prior to Visit 1 | 0.009 | 0.70 | 0.009 | 0.72 | N/A | N/A |
| Antibiotics between Visit 1 and 2 | 0.009 | 0.65 | 0.009 | 0.64 | N/A | N/A |
| **GUT** | | | | | | |
| Level of Education[A] | 0.014 | 0.57 | 0.014 | 0.54 | 0.014 | 0.53 |
| Prenatal Insurance[B] | **0.025** | **0.01** | **0.025** | **0.009** | **0.025** | **0.01** |
| Marital/Cohabiting[C] | 0.014 | 0.65 | 0.014 | 0.60 | 0.014 | 0.63 |
| Antibiotics prior to Visit 1 | 0.016 | 0.34 | 0.016 | 0.35 | N/A | N/A |
| Antibiotics between Visit 1 and 2 | 0.013 | 0.67 | 0.013 | 0.69 | N/A | N/A |

**Notes.**
[1] Model controlling for age, any previous term birth, any previous preterm birth, weeks between sampling.
[2] Model controlling for age, any previous term birth, any previous preterm birth, weeks between sampling, and antibiotics.
[A] Education coded as 1 for high school or lower, 0 for some college or higher.
[B] Prenatal insurance coded as 1 for Medicaid, 0 for Private.
[C] Marital/Co-habiting coded as 1 for Single or Not cohabiting, 0 for Married or Cohabiting.
The bold styling indicates a *p*-value that is statistically significant at $\alpha < 0.05$.

Given that we found some measures of socioeconomic status—which likely reflect access to resources that shape exposures to the physical, social, and psychosocial environments (*Dowd & Renson, 2018*)—associate with the longitudinal stability of the microbiota in pregnancy, and this association is only minimally attenuated by prenatal antibiotic exposure, our next steps will include evaluating whether specific health exposures and behaviors that associate with measures of socioeconomic status, such as stress, dietary intake and nutritional status, and substance use, affect the compositional stability of the microbiota and the influence these effects may have on preterm birth. We will also evaluate whether specific infectious or antibiotic exposures attenuate or modify these associations. As with measures of socioeconomic status, researchers studying the microbiota have captured limited information on psychosocial, cultural, and behavioral factors as well as diet in ancestrally diverse study populations, which may contribute to compositional differences in the microbiota observed by race/ethnicity and/or be linked with health outcomes under study (*Dowd & Renson, 2018*; *Findley et al., 2016*).

## STRENGTHS AND LIMITATIONS

Strengths of this study include its characterization of participating women in terms of indicators of socioeconomic status and antibiotic exposures occurring before and between microbiota sampling, which allowed for the analysis of the effect of these important yet often overlooked exposures on the stability of the microbiota at multiple body sites across pregnancy. A strength of this analysis is its exclusive focus on African American women, allowing for the within-race discernment of whether socioeconomic status influences the stability of the microbiota. This is important given the confounding that often occurs when comparing health exposures and outcomes for women belonging to different racial/ethnic groups that have different distributions of socioeconomic status and factors linked to socioeconomic status.

However, this study is not without limitations. First and foremost, as with any study involving 16S rRNA gene sequencing, the taxonomic assignment of sequences was limited in resolution by the short read length (250 bp paired ends). Also, as with other 16S rRNA gene sequencing studies, the taxonomic assignment and resolution is influenced by the particular primer set selected for amplifying the variable regions of the 16S rRNA gene, which may limit comparability with studies based on other primer sets (*Bukin et al., 2019*). Second, although 110 women is a relatively large cohort compared to many earlier studies, it is not large enough to detect small effect sizes. Nor was this sample size adequate for us to evaluate whether other health exposures and behaviors that associate with measures of socioeconomic status affect the stability of the microbiota in pregnancy. Furthermore, the analyses presented in this exploratory study should be interpreted with caution given that analyses were not adjusted for multiple comparisons (*Althouse, 2016*). Third, the assessment of socioeconomic status by level of education and prenatal health insurance type (Medicaid, private) is somewhat limited and does not fully capture dimensions of socioeconomic status such as employment status and type, family structure or size, that likely influence access to relevant resources and health behaviors (such as dietary patterns, housing conditions) that may influence microbiota composition. This is especially important considering that approximately 75% of the cohort was covered by Medicaid, which could limit our ability to assess the true effect of health insurance type, and the mean age of our cohort was approximately 24 years, which might limit the possible effect of education.

## CONCLUSIONS

For the vaginal site, a low level of education was associated with an increase in Shannon and Chao1 diversity over pregnancy, with minimal attenuation of this relationship by prenatal antibiotic exposure; however, level of education and prenatal insurance type did not affect the longitudinal stability or direction of compositional change of the vaginal microbiota. In contrast, for the oral and gut sites, level of education and prenatal insurance type were not associated with change in measures of alpha-diversity over pregnancy; however, a low level of education and prenatal antibiotic use did affect longitudinal stability of the oral and gut microbiota while only prenatal insurance status was associated with the direction of change in the composition of the oral and gut microbiome. Thus, prenatal antibiotics resulted

in lower stability in the composition of the oral and gut microbiota across pregnancy compared to women who did not receive prenatal antibiotics, but the receipt of prenatal antibiotics did not predict the direction of change in microbiota composition across pregnancy, whereas having prenatal Medicaid compared to private insurance did. In conclusion, our findings in this exploratory analysis of relatively small sample size support that measures of socioeconomic status, including level of education and prenatal insurance type, are variably associated with changes in microbiota alpha- and beta-diversity across pregnancy for the vaginal, oral, and gut body sites. Additional dedicated studies on a larger sample size are needed to confirm these results.

## ACKNOWLEDGEMENTS

The authors are grateful to the women who generously agreed to participate in this research, to the research coordinators who interface with the participating women to carefully collect research data, and to the clinical providers, nursing and laboratory staff at the prenatal care clinics of Grady Memorial Hospital and Emory University Hospital Midtown without whose cooperation this research would not be possible. The findings and conclusions in this report are those of the author(s) and do not necessarily represent the official position of the Centers for Disease Control and Prevention.

### Funding

This study was supported by the National Institutes of Health, National Institute of Nursing Research [R01NR014800], National Institute on Minority Health and Health Disparities [R01MD009064], National Institute of Environmental Health Sciences [R24ES029490] and the Office of the Director [UG3OD023318/UH3OD023318]. This study was also supported in part by the Emory Integrated Genomics Core, which is subsidized by the Emory University School of Medicine and is one of the Emory Integrated Core Facilities. Additional support was provided by the National Center for Advancing Translational Sciences of the National Institutes of Health under Award Number UL1TR000424 and UL1TR000454. The funders had no role in study design, data collection and analysis, decision to publish, or preparation of the manuscript.

### Grant Disclosures

The following grant information was disclosed by the authors:
National Institutes of Health.
National Institute of Nursing Research: R01NR014800.
National Institute on Minority Health and Health Disparities: R01MD009064.
National Institute of Environmental Health Sciences: R24ES029490.
Office of the Director: UG3OD023318/UH3OD023318.
Emory Integrated Genomics Core.
National Center for Advancing Translational Sciences of the National Institutes of Health: UL1TR000424, UL1TR000454.

## Competing Interests

Timothy Read is an Academic Editor for PeerJ. The other authors declare that they have no competing interests.

## Author Contributions

- Anne L. Dunlop and Elizabeth J. Corwin conceived and designed the experiments, performed the experiments, contributed reagents/materials/analysis tools, authored or reviewed drafts of the paper, approved the final draft.
- Anna K. Knight, Glen A. Satten and Anya J. Cutler analyzed the data, prepared figures and/or tables, authored or reviewed drafts of the paper, approved the final draft.
- Michelle L. Wright and Rebecca M. Mitchell analyzed the data, authored or reviewed drafts of the paper, approved the final draft.
- Timothy D. Read and Jennifer Mulle conceived and designed the experiments, performed the experiments, authored or reviewed drafts of the paper, approved the final draft.
- Vicki S. Hertzberg analyzed the data, contributed reagents/materials/analysis tools, authored or reviewed drafts of the paper, approved the final draft.
- Cherie C. Hill performed the experiments, authored or reviewed drafts of the paper, approved the final draft.
- Alicia K. Smith conceived and designed the experiments, contributed reagents/materials/analysis tools, prepared figures and/or tables, approved the final draft.

## Human Ethics

The following information was supplied relating to ethical approvals (i.e., approving body and any reference numbers):

The research protocol was reviewed and approved by the Emory University Institutional Review Board (protocol number 68441).

## Data Availability

De-identified raw sequence data is available at the NCBI Short Read Archive: SRX012800.

Phenotype and sequence data is available at dbGaP: https://www.ncbi.nlm.nih.gov/projects/gap/cgi-bin/study.cgi?study_id=phs001865.v1.p1.

Raw sequence data is available at Zenodo: Dunlop, Anne, Corwin, Betsy, Knight, Anna, & Smith, Alicia. (2019). Emory University African American Microbiome in Pregnancy [Data set]. Zenodo. http://doi.org/10.5281/zenodo.3341723.

## Supplemental Information

Supplemental information for this article can be found online at http://dx.doi.org/10.7717/peerj.8004#supplemental-information.

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
