# Peer review of "Stability of the vaginal, oral, and gut microbiota across pregnancy among African American women: the effect of socioeconomic status and antibiotic exposure"

_PeerJ, doi:10.7717/peerj.8004_

## Round 0.1 · original submission · Minor Revisions

Your article has been reviewed by two experts, and based on their reports I am recommending you undertake minor revisions of its content. Please address all reviewers' comments when revising the manuscript, particularly those relating to the Introduction. I look forward to receiving your revised manuscript in due course.

Reviewer 1 ·

Basic reporting

This manuscript reports on the stability of the vaginal, oral, and gut microbiota across pregnancy in African American women and examines the effect that socioeconomic status and antibiotic exposure has.

The use of the term ‘microbiome’ in the context of this manuscript is not appropriate. ‘Microbiota’ should be used instead as this refers primarily to the analysis of 16S rRNA genes. See (Marchesi and Ravel., 2015. Microbiome 3:31) which provides an adequate definition of both terminologies.

Specific comments which relate to each section of this manuscript;

ABTRACT
- Clear aims listed, with professional English used throughout manuscript.
- In Line 43 you mention Shannon and Chao1 diversity and Bray-Curtis dissimilarity but do not specify that the former two are related to alpha diversity and the latter refers to beta-diversity. Please highlight this here instead of mentioning it in the results section.
- The conclusion is simply a repeat of the results and does not include any closing statements.

INTRODUCTION
- It would be helpful for the reader if you define what the microbiota is to provide clearer context.
- The papers referenced on Line 63 (1 to 3) are not appropriate for the opening statement of this introduction. I suggest looking at microbial functions at different body sites e.g. gastrointestinal, vaginal and so forth for more appropriate references e.g. (Round and Mazmanian, 2009. Nat Rev Immuno, 9 (313-323). There is also a missing full stop after this sentence.
- Again, look at the literature for more appropriate studies that have explored the role of the microbiota in infection, digestion, stress and so on (Line 64, 65 and 66).
- On Line 70/71 you say [there have been...] “few studies that have investigated the role of the microbiome of other body sites during pregnancy.” but do not list them please provide them.
- Please provide references to the few studies you refer to on Line 72 to 75.
- 16S rRNA gene sequencing is not used to study the microbiome it is used to study the microbiota. Please make changes to this term (Line 77).
- There is no mention of the defined community state types that are described in reference 26 (Ravel et al., 2011. Proc Natl Acad Sci U S A). Please describe them here. Results from MacIntyre et al., 2015. Sci Rep 5(8988) can also be discussed in this section.
- In Line 86 you refer to L. iners as a species that produces less acid – are you making a point about pH here? If so, please elaborate or remove as it is a redundant comment.
- In Line 89 you say little has been published in terms of evaluating the factors which contribute to differences in microbiome composition, but you have failed to mention studies that have looked specifically at racial/ethnic differences in general e.g. (Deschasaux et al., 2018. Nat Med. 24 (1526-1531)), (Brooks et al., 2018. PLoS Biol. 16(12): e2006842)) etc.
- In Line 96 the paper (ref 31) correctly refers to the microbiota and not microbiome, please change.
- Please highlight/re-iterate at the end of Line 98 why your study is important.
- References needed for the sentence that finishes on Line 101.
- sp. and spp. are not italicised (Line 119 and 122)
- The sentence starting on Line 132 to 137 should be re-worded as it currently doesn’t make sense. You don’t use “16s rRNA gene sequencing of the microbiome”, you carry out 16S rRNA sequencing in samples (also written on Line 138).
- Please include the most recent papers with regards to bacteria in the placenta and amniotic fluid or mention the fact these may be related to contamination (Line 143). There is a plethora of articles that have explored the presence of bacteria (or lack of) in the placenta over the last few years.
- Please re-examine the literature with regards to your statement on Line 144/145 about there being few studies that have investigated the gut microbiota during pregnancy.
- Provide references for the Sociodemographic features and prenatal microbiota section as none are given.
- List the studies mentioned in Line 172/173.

METHODS
- The metadata/raw data files have been opened with no problems encountered.
- Figures and tables conform to guidelines.
- The sentence starting on Line 314 should be revised as it does not currently read well.
- The word “differences” is inappropriately italicised on line 324.

RESULTS
- In line with typical results section formatting it would be more appropriate to introduce your participants at the start (Line 341 onwards) followed by a brief overview of the 16S rRNA sequencing data and then start a new paragraph which presents the sequencing results.
- With regards to the 16S rRNA sequencing data section please clarify in more detail which samples were removed because it currently doesn’t read well.
- The p values presented in Line 359 and 361 do not have a reference (i.e. which Tables they are derived from)

Experimental design

Specific comments which relate to each section of this manuscript;

INTRODUCTION
- The goals of this study are clear and well defined (Line 181 to 193). Please consider using the common acronym, SES, for socioeconomic status.

METHODS
- The participants were selected from the Emory University African American Vaginal, Oral, and Gut Microbiome in Pregnancy Study (ref 53) which is a robust and well-designed study.
- The protocol review number is provided and has been viewed in the supplementary file. No issues found.
- The DNA extraction methodology employed here is robust and is easily reproducible. The V3-V4 primers are fine but please be aware of primer limitations (this should be mentioned in the discussion).
- Please clarify in the section related to DNA extraction, 16S rRNA gene library preparation and sequencing if a negative control/kit control was run as it is not mentioned. If this was not, provide justification as to why this did not occur (note that not running controls is not good practice). Several papers over the last few years provide means of dealing with kit contamination e.g. (Sheik et al., 2019. Front Microbiol. 9: 840).
- The use of Qiime2 and dada2 is commended here.
- Please clarify which software you used to perform the statistical analysis and include software version number.
- Please provide reasoning behind comparing women who provided paired samples with those that didn’t (Line 285/286). This comparison is atypical and would normally involve a comparison between the women that provided paired samples and an independent comparison comparing those that did not contribute paired samples.

Validity of the findings

Specific comments which relate to each section of this manuscript;

DISCUSSION
- In general, this discussion would benefit from being better contextualised with the relevant literature.
- The opening statement of the discussion should provide a brief overview of the authors findings before drawing conclusions and findings from other relatable studies.
- Ensure all studies are appropriately referenced through text.
- The limitations of 16S rRNA sequencing should be better contextualised. It would also be noteworthy to mention primer design and contamination influences.
- There appears to be no speculation on why these findings occurred. Please comment as appropriate.

Additional comments

STRENGTHS
- This study addresses a literature gap of how the microbiota changes across body site after accounting for SES and antibiotic exposure.

WEAKNESSES
- The literature review is not sufficient and does not provide enough or appropriate references. This manuscript would also benefit from being better contextualised in the discussion which is currently oversimplified with limited interpretation of the data.
- Given the plethora of data generated from the 16S rRNA sequencing carried out it is surprising to see that taxonomy is not covered at all e.g. dominant genera found across body site. Moreover, there is the capacity to generate a number of very interesting graphs using the Phyloseq package that was employed for data analysis.

If this manuscript is to be recommended for PeerJ publication, all points made must be addressed.

Reviewer 2 ·

Basic reporting

This manuscript is quite well-written, well-referenced, well-structured and self-contained in the context of the state of the field.

Experimental design

The manuscript, which examines the correlations of the oral, vaginal and rectal microbiomes across pregnancy in women of varied socioeconomic status (etc.) seems to fit very well the Aims and Scope of this journal. The research questions ["to investigate: (1) the stability of the vaginal, oral, and gut microbioime from early (8-14 weeks) through later (24-30 weeks) pregnancy among African American women according to measures of socioeconomic status, accounting for prenatal antibiotic use; (2) whether measures of socioeconomic status are associated with changes in microbiome composition over pregnancy; and (3) whether exposure to prenatal antibiotics mediate any observed associations between measures of socioeconomic status and stability of the vaginal, oral, and gut microbiome across pregnancy] are well defined and have the potential to add to the overall state-of-the-field. There is no concern about the technical or ethical standards, and the methods are described sufficiently, with a single caveat. Line 260 states: "V3 and V4 regions of the 16S rRNA gene were amplified and tailed using target specific primers", citing a 2011 manuscript. It is not clear if these investigators used these primers, nor is their a discussion of the appropriateness of these primers for vaginal samples. Overall, this is a minor concern that should be readily addressed by the authors.

Validity of the findings

The data presented are appropriate. Filtering of the data was done according to rigorous and well-described standards. The results complement data from multiple other publications, but do not directly repeat any to any substantial degree. There is merit to the publication of these data. The data presented and the analysis of it are appropriate. The analysis is unique in there is almost no reference to specific taxa. The analysis and conclusion are based entirely on diversity (Shannon and Bray-Curtis) measures. Normally, there would be a reference to taxa present, absent, or changing in predominance across pregnancy. I would not suggest that the manuscript be refocused for this, but it is unusual. Overall, the data seem to be robust, and the statistical analysis sound.

The conclusions are well-expressed, not over-interpreted, and significant for the field. SES (and race) have been overlooked as variables in many publications and these variables seem to have a larger than expected effect. The conclusion that antibiotics seem to have no great impact is a very interesting and somewhat unexpected finding. The authors do not expand on this, but some thoughts about why this may be might be worthwhile adding to the manuscript.

Additional comments

This is a sound body of work and adds to the gestalt of our understanding of the correlations of the microbiome diversity (and composition) to the changes going on during pregnancy.

---

## Round 0.2 · accepted · Accept

Thank you for revising your manuscript in line with reviewer 1's comments. I look forward to seeing your article published.